# A Multi-Trait Association Analysis of Brain Disorders and Platelet Traits Identifies Novel Susceptibility Loci for Major Depression, Alzheimer’s and Parkinson’s Disease

**DOI:** 10.3390/cells12020245

**Published:** 2023-01-06

**Authors:** Alfonsina Tirozzi, Miriam Shasa Quiccione, Chiara Cerletti, Maria Benedetta Donati, Giovanni de Gaetano, Licia Iacoviello, Alessandro Gialluisi

**Affiliations:** 1Department of Epidemiology and Prevention, IRCCS Neuromed, Via dell’Elettronica, 86077 Pozzilli, Italy; 2EPIMED Research Center, Department of Medicine and Surgery, University of Insubria, Via Ravasi 2, 21100 Varese, Italy

**Keywords:** Alzheimer’s disease, Parkinson’s disease, major depressive disorder, genomics, multi-trait associations, platelets

## Abstract

Among candidate neurodegenerative/neuropsychiatric risk-predictive biomarkers, platelet count, mean platelet volume and platelet distribution width have been associated with the risk of major depressive disorder (MDD), Alzheimer’s disease (AD) and Parkinson’s disease (PD) through epidemiological and genomic studies, suggesting partial co-heritability. We exploited these relationships for a multi-trait association analysis, using publicly available summary statistics of genome-wide association studies (GWASs) of all traits reported above. Gene-based enrichment tests were carried out, as well as a network analysis of significantly enriched genes. We analyzed 4,540,326 single nucleotide polymorphisms shared among the analyzed GWASs, observing 149 genome-wide significant multi-trait LD-independent associations (*p* < 5 × 10^−8^) for AD, 70 for PD and 139 for MDD. Among these, 27 novel associations were detected for AD, 34 for PD and 40 for MDD. Out of 18,781 genes with annotated variants within ±10 kb, 62 genes were enriched for associations with AD, 70 with PD and 125 with MDD (*p* < 2.7 × 10^−6^). Of these, seven genes were novel susceptibility loci for AD (EPPK1, TTLL1, PACSIN2, TPM4, PIF1, ZNF689, AZGP1P1), two for PD (SLC26A1, EFNA3) and two for MDD (*HSPH1*, *TRMT61A*). The resulting network showed a significant excess of interactions (enrichment *p* = 1.0 × 10^−16^). The novel genes that were identified are involved in the organization of cytoskeletal architecture (*EPPK1*, *TTLL1*, *PACSIN2*, *TPM4*), telomere shortening (*PIF1*), the regulation of cellular aging (*ZNF689*, *AZGP1P1*) and neurodevelopment (*EFNA3*), thus, providing novel insights into the shared underlying biology of brain disorders and platelet parameters.

## 1. Introduction

Platelets have represented, for decades, an interesting setting to investigate the biological underpinnings of neuropsychiatric and neurodegenerative disorders since they are considered “circulating mirrors of neurons” [1]. Indeed, despite their different embryonic origin, platelets and neurons share common characteristics in subcellular organization and in protein composition. There are proteins typically expressed in both neurons and circulating platelets, and they were found to regulate processes such as platelet activation, hemostasis and thrombosis [2]. For example, Reelin- neuronal protein that regulates cell migration, synaptic plasticity and memory formation- is also expressed in blood and is actively released following platelet activation [3,4]. Amyloid Aβ peptides, which accumulate in senile plaques in dementia, and the amyloid precursor protein (APP), are expressed in megakaryocytes, stored in platelet α-granules and released upon platelet activation [1]. Released APP is able to participate in hemostasis and can trigger platelet activation, adhesion and aggregation through a number of different pathways [5]. Soluble APP inhibits the activity of the blood coagulation factors IXa, XIa and Xa, and, to a lesser extent, of factor VIIa–tissue factor complex [6]. It plays a role in the coagulation cascade, modulating hemostasis following vascular injury. Brain derived neurotrophic factor (BDNF)-a secretory protein regulating the development and function of neural circuits-is expressed in both central and peripheral nervous systems but also in human megakaryocyte α-granules together with platelet factor 4 (PF4), where they are stored and released by platelets at the site of injury during platelet aggregation [5,7]. Similarly, serotonin—a neurotransmitter with important roles in controlling behavior and sociality—is stored in platelet-dense granules, where it is released upon activation to act as a weak agonist [3,4].

Epidemiological and genetic evidence also supports the existence of a bridge between platelet traits and neurodegenerative/neuropsychiatric disorders, such as Alzheimer’s disease (AD), Parkinson’s disease (PD) and major depressive disorder (MDD), which often suffer from a lack of risk-predictive biomarkers [5]. Among candidate circulating biomarkers, platelet parameters such as platelet count (Plt), mean platelet volume (MPV) and platelet distribution width (PDW) have been associated with the risk of MDD, AD and PD [7]. Some epidemiological studies consistently reported a positive association between MPV and MDD [5,7,8,9,10], as well as between depressive symptoms and PDW, while the evidence of an association between Plt and MDD status is less consistent [7]. Similarly, increased MPV was also reported for PD-although other studies observed an inverse correlation with the staging of the disease [7,11]-and with AD and cognitive performance, although not always consistently across studies and types of dementia [7]. PDW has been instead more consistently (inversely) associated not only with AD risk [5] but also with mild cognitive impairment and vascular dementia [12].

More recently, genomic studies investigated the genetic correlation-or single nucleotide polymorphism (SNP)-based co-heritability-of platelet traits and brain disorders, as well as potential causality links through Mendelian randomization (MR) approaches. In a large genome-wide association study (GWAS) of blood cell measures, Astle et al. performed a multivariable MR analysis on platelet parameters and MDD risk, which revealed no significant causal effect of the formers on the latter, although MPV and PDW showed marginally significant effects [5]. Later on, Wray et al. investigated genetic correlations between MDD risk and Plt and MPV, reporting no significant genetic correlations between depression and platelet parameters [13], which we instead observed for PDW and MDD risk [14]. Our group later detected a significant genomic overlap between PDW and PD risk through linkage disequilibrium (LD) score regression, which was confirmed also by a polygenic score analysis, as well as a trend of significance for genetic correlations between PDW and AD risk [5]. Despite these promising genetic findings, the variants and genes at the basis of the genomic overlap between platelet parameters and brain disorders remain largely under-investigated. A first attempt was made to carry out a multi-trait genetic association analysis between PD age at onset and MPV, which revealed novel associations in interesting candidate genes such as *KALRN* (Kalirin RhoGEF Kinase), encoding a PINK1 interactor previously implicated in schizophrenia, AD and PD [5].

Here, we aimed to identify novel genetic associations with AD, PD and MDD using a multi-trait association analysis (MTAG) approach [15], exploiting their reciprocal genetic correlations and evidence of genomic overlap previously identified among themselves and with three common platelet parameters, namely, Plt, MPV and PDW. Such an approach provided higher power to detect novel susceptibility variants for neuropsychiatric and neurodegenerative disorders, as well as overlapping genes and pathways enriched for these associations. This allowed us to untangle the resulting molecular networks at a more fine-grained resolution, revealing potential molecular targets for future treatments of these disorders.

## 2. Materials and Methods

We used MTAG [15] to perform a multi-trait association analysis of three neuropsychiatric/neurodegenerative disorders-AD, PD and MDD-and three common platelet parameters that have been associated with their risk, namely, Plt, MPV and PDW [7]. MTAG is a generalization of standard, inverse-variance-weighted meta-analysis between two or more genetically correlated traits, which generates trait-specific associations for each genetic variant shared among the source GWAS studies, using linkage disequilibrium (LD) score regression to account for potential sample overlap [15]. The summary statistics of AD (71,880 cases, 383,378 controls and 13,309,438 SNPs) [16], PD (37,688 cases, 18,618 UK Biobank proxy-cases, 1,417,791 controls and 10,081,487 SNPs) [17], MDD (246,363 cases, 561,190 controls and 7,880,531 SNPs) [18], Plt (38,561,936 SNPs), MPV (41,254,093 SNPs) and PDW (41,253,708 SNPs; all with N = 408,112 participants of European ancestry) [19] were obtained from published GWAS summary statistics data (see URLs). To carry out MTAG analysis, we first pre-processed and quality-controlled summary statistics from each study involved: SNP (rs) ids were retrieved, and Z-score was computed by log (OR)/SE when these were not available; MAF threshold was set at 0.01; indels were removed and SNPs mapping to the same position of other variants and/or showing conflicting alleles among different studies were dropped. After variant filtering, the number of variants left for analysis was 8,632,257 for AD; 6,582,074 for PD; 7,183,400 for MDD; 8,933,201 for Plt; 8,933,763 for MPV; and 8,934,170 for PDW. Of these, 4,540,326 variants were in common among the different studies and, therefore, underwent MTAG analysis. Using Functional Mapping and Annotation (FUMA) platform [20], we first identified LD-independent genome-wide significant SNPs within each study (*p* < 5.0 × 10^−8^; pairwise r^2^ < 0.6 in a 1 Mb window). Then, we selected novel genome-wide significant SNP associations with the analyzed disorders, defined as associations not detected as genome-wide significant in the original study, nor in previous GWASs of the same disorder, based on the GWAS catalog and literature search until 30 April 2022 (Table 1).

To have further biological insights into the underlying biology of common variance in different neuropsychiatric/neurodegenerative disorders and platelet parameters, associations underwent gene, gene ontology and pathway enrichment analysis through MAGMA v1.08 [21], within the FUMA platform [20]. This was carried out for all the protein-coding genes to which at least one SNP was annotated within a ±10 kb interval, namely, 18,781 genes. A Bonferroni correction for multiple testing was applied accordingly, based on the number of genes tested (α = 2.7 × 10^−6^) (Table 2).

To estimate protein–protein interactions (PPIs) among the genes enriched for associations, we used the search tool for the retrieval of interacting genes/proteins in db-STRING v11.5 [22]. We analyzed genes significantly enriched for associations with AD, PD and MDD, first separately and then merged into a single list, to compute “global” interactions among all the genes significantly enriched for any of the three disorders. The obtained network included both direct (physical) and indirect (functional) associations, specifically evidence of interaction from curated databases; evidence experimentally determined; gene neighborhood; gene fusions; gene co-occurrence; joint gene mentioning based on text mining in published articles; gene co-expression; and protein homology. We set a minimum interaction score of >0.7 so that only high-confidence interactions between proteins were included in the analysis. The observed excess of interactions compared to the expected number of edges among nodes, average node degree (i.e., the average number of edges per node in the graph) and local clustering coefficient (i.e., a measure of the extent to which nodes in the graph tend to cluster together) were taken as measures of network density and clustering levels.

## 3. Results

In the multi-trait association analysis of three different brain disorders (AD, PD, MDD) and three different platelet parameters (Plt, MPV and PDW), we analyzed 4,540,326 variants shared across the different source studies that passed QC (see Methods section). Given the main aim of the manuscript, we report in detail below the results of the associations with AD, PD and MDD, along with the overlap with platelet parameters in terms of single variant and the gene enrichment of associations. The full results of the MTAG analysis of platelet traits are reported in Appendix A (Appendix A) and online (see Data Availability Statement). MTAG revealed 358 genome-wide significant multi-trait associations (*p* < 5 × 10^−8^; pairwise r^2^ < 0.6; Figure 1): 149 for AD (top hit at rs1081105, within the APOE gene, *p* = 2.36 × 10^−180^), 70 for PD (top hit at rs356219, within the SNCA gene, *p* = 6.01 × 10^−41^) and 139 for MDD (top hit at rs2232429, within the ZSCAN12 gene, *p* = 6.41 × 10^−19^). Among these, we observed 27 novel associations with AD (top hit at rs2232429, within the *ITGB5* gene, *p* = 6.41 × 10^−19^), 34 with PD (top hit at rs1372518 within the *SNCA* gene, *p* = 3.33 × 10^−28^) and 40 with MDD (top hit at rs200965 in a transcription factor binding site on 6p22.1, *p* = 4.07 × 10^−15^). Of these SNPs, seven were mapped in novel susceptibility genes for AD (*EPPK1*, *TTLL1*, *PACSIN2*, *TPM4*, *PIF1*, *ZNF689* and *AZGP1P1*), two for PD (*SLC26A1* and *EFNA3*) and two for MDD (*HSPH1* and *TRMT61A*). The analysis of 18,781 genes tested in the gene-based enrichment analysis revealed 62, 70 and 125 genes with a significant enrichment of associations with AD, PD and MDD, surviving correction for multiple gene testing (*p* < 2.7 × 10^−6^, Figure 2). Of these, eight genes represented novel associations with AD (*PVRIG*, *FAM57B*, *NDFUS2*, *C16orf92*, *KLC3*, *B4GALT3*, *ZNF688* and *DEDD*), 14 with PD (*DPM3*, *SLC26A1*, *FBXL19*, *SMIM15*, *ERCC8*, *FAM200B*, *CTF1*, *ADAM15*, *PRSS8*, *NCOR1*, *VKORC1*, *ZNF668*, *PRSS36* and *SRCAP*) and 38 with MDD (*ZNF165*, *BTN2A2*, *OR2W1*, *OR12D3*, *TRMT61A*, *OR2J1*, *HMGN4*, *OR2B3*, *BTN3A3*, *ZNF197*, *ZNF35*, *TRIM27*, *OR2B6*, *DLST*, *RHOBTB1*, *ZNF660*, *RBM4B*, *ITGB6*, *RBM14-RBM4, HIST1H4K, HIST1H2AK, HIST1H2BL, HIST1H3H, HIST1H2AL, HIST1H1B, HIST1H3I, HIST1H4J, HIST1H2BM, HIST1H2AI, HIST1H2AJ, HIST1H2AM, HIST1H3J, HIST1H2BN, HIST1H2AG, HIST1H2BO, HIST1H4L, HIST1H2BJ* and *HIST1H4I*). None of these genes were overlapping across the three disorders. However, when we checked the overlap between each neurodegenerative/neuropsychiatric disorder and platelet parameters in terms of single variant associations, we found 24 SNPs associated with AD and 3 associated with PD, which were also associated with 1 or more platelet parameters. Of these, 12 AD- and 3 PD-associated SNPs represented novel associations never detected before (Table 3a,b). As for overlapping gene enrichments with any of the platelet parameters analyzed, we identified 29 genes matching with AD, 14 with PD and 15 with MDD. Of these, 6 were novel susceptibility loci for AD, 4 for PD and 12 for MDD (Table 4a–c).

The molecular network resulting from the gene enrichment test, as produced by STRING v11.5 analysis, showed more significant interactions than expected for AD (enrichment *p* = 1.0 × 10^−16^; 57 nodes, 57 edges vs. 2 expected, average node degree 2.00 and average local clustering coefficient 0.36), PD (enrichment *p* = 8.46 × 10^−9^; 63 nodes, 18 edges vs. 3 expected, average node degree 0.57 and average local clustering coefficient 0.23) and MDD (enrichment *p* = 1.0 × 10^−16^; 120 nodes, 264 edges vs. 49 expected, average node degree 4.4 and average local clustering coefficient 0.47) (Appendix A; Appendix A). Similarly, we observed evidence of an interaction also when the genes enriched for the three disorders were analyzed together (enrichment *p* = 1.0 × 10^−16^; 240 nodes, 281 edges vs. 61 expected, average node degree 2.34 and average local clustering coefficient 0.3) (Figure 3; Appendix A).

Gene-set analysis also revealed significant enrichment for the three disorders, the most associated gene ontology (GO) term was *negative regulation of amyloid precursor protein catabolic process* (13 genes, β(SE) = 2.23(0.26); enrichment *p* after Bonferroni correction = 2.7 × 10^−13^) for AD, *IgG binding* (9 genes, β(SE) = 0.03(0.31); P_bonf_ = 0.0056) for PD and *GABAergic synapse* (64 genes, β(SE) = 0.74(0.13); P_bonf_ = 9.5 × 10^−5^) for MDD (Appendix A; see URLs to access the full list of pathways tested and the genes driving these enrichments).

## 4. Discussion

We report the first multi-trait association analysis of structural platelet parameters routinely assessed in blood tests and three of the most common neurodegenerative/neuropsychiatric disorders, identifying novel candidate susceptibility genes for AD, PD and MDD. The most significant associations were detected in some of the most implicated genes in neurodegenerative/neuropsychiatric disorders, namely, *APOE* (apolipoprotein E, with AD) [39], *SNCA* (alpha synuclein, with PD) [40] and *ZSCAN12* (zinc finger and SCAN domain-containing 12, with MDD) [41]. APOE is a protein associated with lipid particles that mainly functions in lipoprotein-mediated lipid transport between organs via plasma and interstitial fluids [42]. Alpha synuclein is involved in synaptic activities such as the regulation of synaptic vesicle trafficking and subsequent neurotransmitter release [43,44]; moreover, it modulates DNA repair processes, including the repair of double-strand breaks [45]. *ZSCAN12* encodes a Zinc finger and SCAN domain-containing protein involved in transcriptional regulation.

Still, MTAG analysis also revealed novel genes showing significant multi-trait associations, which, to our knowledge, were never associated with these disorders before. Among genes associated with AD, *EPPK1*, *TTLL1*, *PACSIN2* and *TPM4* play a role in the organization of cytoskeletal architecture, which has been identified as an important component in the development of neurodegenerative disorders [46,47,48,49]. *PIF1* prevents telomere elongation by inhibiting the action of telomerase, while *ZNF689* and *AZGP1P1* are transcription factors involved in cell viability and apoptosis, and both molecular functions can affect cellular aging and the development of age-related disorders such as AD and PD [24]. Moreover, *TTLL1* [19], *PACSIN2* [26,40], *TPM4* [27] and *PIF1* [19] were previously associated with platelet parameters, suggesting a possible pleiotropic effect of these genes. *EFNA3*, a novel gene resulting in an association with PD, encodes a member of the ephrin family, previously implicated in mediating developmental events, especially in the central nervous system [50].

A gene-based enrichment analysis also revealed novel genes associated with AD, PD and MDD. Interestingly, among these genes are several encode transcription factors that may be involved in development, maintenance and survival of neurons and olfactory receptors [51]. Indeed, olfactory dysfunction, which is thought to be due to the loss of synaptic function, has been linked with most neurodegenerative, neuropsychiatric and communication disorders [52]. Moreover, among these genes, there are also some histone complex proteins, in line with some recent studies revealing associations between histone methylation/acetylation and AD [53] and implicating several histone deacetylases in the pathogenesis of PD [54]. These findings suggest the pleiotropic influence of several genes on the risk of neurodegenerative and neuropsychiatric disorders, which were not previously detected through classical univariate GWAS analyses.

Of note, we found several overlaps between genes and SNPs multi-trait associations with the brain disorders and platelet parameters analyzed. Among genes enriched for association, we identified clusters of genes encoding products involved in mitochondrial function (e.g., *NDUFS2*, *NDUFAF2* and *TOMM40L*), cytoskeleton remodeling (*CD2AP* and *KLC3*, as discussed above) and histone proteins (*HIST1H2BK*, *HIST1H4K, HIST1H2AK* and *HIST1H3B*, as explained below). Indeed, *NDUFS2* and *NDUFAF2* encode for a subunit and for a chaperone involved in the assembly of complex I, located on the inner mitochondrial membrane, while *TOMM40L* is involved in mitochondrial transmembrane translocation. Several studies suggest that platelet mitochondrial dysfunction may be involved in neurodegenerative diseases such as AD and PD [55,56,57]. Still, further studies are needed to clarify the variant association overlap between platelet parameters and MDD, which we were not able to identify here, possibly due to the genetic and phenotypic heterogeneity of depression.

Protein–protein interaction analysis revealed a significant excess of interactions among enriched genes for the brain disorders tested both separately and jointly, suggesting that their gene products are highly likely to be linked in a global molecular network.

In particular, this highlighted some local networks of interests, such as the one among the histone proteins complex-*HIST1H2BI*, *HIST1H2BF* and *HIST1H2BJ*-which may play a role in the onset of neurodegenerative diseases due to the alteration of methylation patterns [58]. Similarly, the apolipoproteins APOE, APOA2, APOC1 and APOC4 have been repeatedly implicated in triglyceride and cholesterol transport and metabolism [59,60], as well as in neurodegenerative [61] and cardiovascular risk [62], while the local network among *EPHA1*, *EPHB2*, *EFNA1*, *EFNA3* and *EFNA4*, highlights the importance of the interactions between ephrins and ephrin receptors in the etiology of several neurodegenerative and neuroinflammatory disorders, suggesting potential links with (cellular) immunity [63].

Gene-set analysis revealed significant enrichments of GO terms involved in the regulation, formation and catabolic processes of amyloid beta and in the negative regulation of metalloendopeptidase activity for AD, supporting the hypothesis that metallopeptidases are implicated in the pathogenesis of several central nervous system diseases such as multiple sclerosis and AD [64]. For PD, significant enrichments of IgG binding is interesting in light of a higher fraction of IgG, a different IgG glycosylation profile [65], and of increased IgG (but not IgM) binding in dopaminergic neurons of PD cases vs. controls [66]. Moreover, serum IgG levels in PD patients are negatively associated with mood/cognition scores [67], in line with a potential pleiotropic role of humoral immunity at the interface among the mood, cognitive and motor control domains. Similarly, the significant enrichment of GABAergic synapse GO term in MDD analysis corroborates the hypothesis that the alteration in GABAergic receptors may play a role in long-term depression [68].

Overall, we provide insights into the shared underlying biology of these disorders and related platelet parameters, proposing novel molecular targets for the risk prediction and treatment of these disorders.

### Strengths and Limitations

The strengths of this study include the novelty of the analysis performed; indeed, to our knowledge, this represents the first attempt to identify the shared genomic underpinnings of platelet parameters and three of the most common neurodegenerative/neuropsychiatric disorders through a comprehensive approach, including not only multi-trait association analysis but also gene-/gene-set enrichment and molecular network analyses. Moreover, our analyses are based on large GWASs, with an important amount of genetic data, which confers robustness to our observations. Last, our focus on novel associations allowed identifying proteins and biological pathways, which should be functionally validated in the future.

This study presents some limitations. First, currently only partial evidence of genetic correlation among the disorders and platelet parameters tested exists, which may have hampered the power of the analyses. Second, in multi-trait association approaches, association significance is often driven by the largest source GWAS involved in the MTAG analysis, which may have biased the analyses towards the largest studies. Still, this represents a useful approach to identifying the pleiotropic variants and genes influencing multiple traits and/or disorders, which were already proven successful with multiple correlated phenotypes [15]. Third, functional studies are warranted to explain the role of the novel susceptibility genes identified here, both in neurodegenerative/neuropsychiatric risk and platelet variability.

However, although there are some limitations, these studies may reveal potential molecular targets for future treatments of three of the most common neurodegenerative/neuropsychiatric disorders.

## Figures and Tables

**Figure 1 cells-12-00245-f001:**
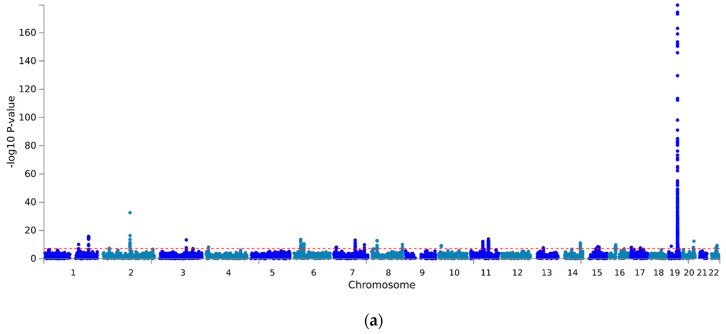
Manhattan plots of multi-trait associations with (**a**) AD, (**b**) PD and (**c**) MDD. The *x*-axis shows chromosomal position, and the *y*-axis shows association *p*-values on a −log10 scale. Red dashed line represents the statistical significance thresholds (α = 5 × 10^−8^).

**Figure 2 cells-12-00245-f002:**
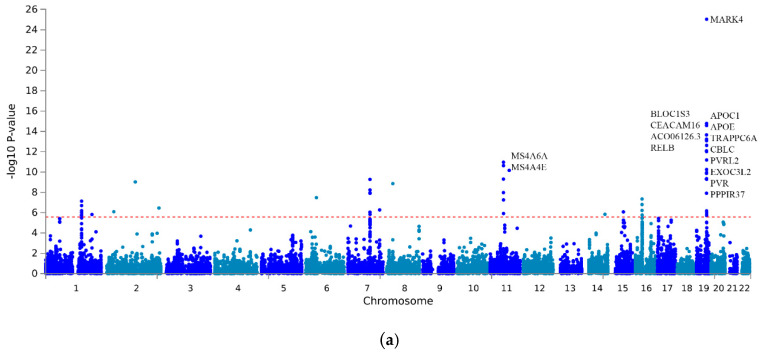
Manhattan plots of gene-based enrichments of multi-trait associations with (**a**) AD, (**b**) PD and (**c**) MDD. The *x*-axis shows chromosomal position, and the *y*-axis shows association *p*-values on a −log10 scale. Red dashed line represents the statistical significance thresholds (α = 2.7 × 10^−6^).

**Figure 3 cells-12-00245-f003:**
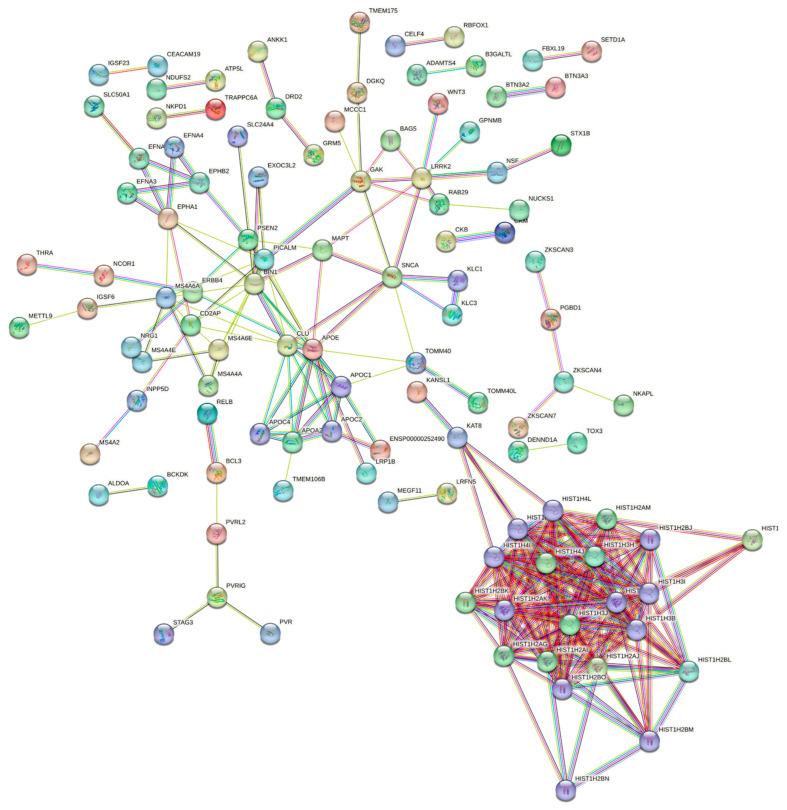
Protein–protein interaction network of genes significantly enriched for associations with AD, PD and MDD. The reported network, including both direct (physical) and indirect (functional) associations, was based on the STRING v11.5 database [22]. Only high-confidence interactions between proteins are reported (interaction score > 0.7), while disconnected nodes in the network are hidden. Each node represents all the proteins produced by a single protein-coding gene locus, while edges represent protein–protein associations. Line color indicates the type of interaction evidence: light blue— from curated databases; purple—experimentally determined; green—gene neighborhood; red—gene fusions; blue—gene co-occurrence; yellow—text mining; black—co-expression; and violet—protein homology.

**Table 1 cells-12-00245-t001:** Novel genes associated with neuropsychiatric/neurodegenerative disorders based on single variant associations.

CHR	Pos	REF	ALT	rsID	*p*	nSNPs	Gene	Disorder
8	144,992,361	C	T	rs7822511	8.82 × 10^−11^	139	EPPK1	AD
22	43,414,330	A	G	rs3091364	3.82 × 10^−10^	16	TTLL1	AD
22	43,279,611	A	G	rs4822218	3.91 × 10^−10^	9	PACSIN2	AD
19	16,211,630	A	G	rs59508494	1.49 × 10^−9^	3	TPM4	AD
15	65,170,949	C	T	rs2013555	6.37 × 10^−9^	72	PIF1	AD
16	30,902,353	A	G	rs80095680	3.62 × 10^−8^	89	ZNF689	AD
7	99,581,469	C	T	rs11761882	4.64 × 10^−8^	34	AZGP1P1	AD
4	975,238	C	T	rs73211813	2.99 × 10^−10^	45	SLC26A1	PD
1	155,053,719	C	T	rs1462855	4.17 × 10^−8^	36	EFNA3	PD
14	104,000,183	C	T	rs2756127	8.20 × 10^−11^	108	TRMT61A	MDD
13	31,733,057	A	G	rs41292151	5.34 × 10^−9^	23	HSPH1	MDD

**Table 2 cells-12-00245-t002:** Novel genes associated with (a) AD, (b) PD and (c) MDD based on gene-based enrichment analysis.

(a)						
SYMBOL	CHR	START	STOP	NSNPS	ZSTAT	*p*
PVRIG	7	99,805,864	99,829,113	43	5.5752	1.24 × 10^−8^
FAM57B	16	30,025,748	30074299	42	5.1135	1.58 × 10^−7^
NDUFS2	1	1.61 × 10^8^	1.61 × 10^8^	43	5.0689	2.00 × 10^−7^
C16orf92	16	30,024,655	30,049,057	27	4.8555	6.01 × 10^−7^
KLC3	19	45,826,692	45,864,778	46	4.6901	1.37 × 10^−6^
B4GALT3	1	1.61 × 10^8^	1.61 × 10^8^	18	4.6514	1.65 × 10^−6^
ZNF688	16	30,570,667	30,594,055	5	4.6452	1.70 × 10^−6^
DEDD	1	1.61 × 10^8^	1.61 × 10^8^	22	4.6293	1.83 × 10^−6^
**(b)**						
**SYMBOL**	**CHR**	**START**	**STOP**	**NSNPS**	**ZSTAT**	** *p* **
DPM3	1	1.55 × 10^8^	1.55 × 10^8^	17	7.3804	7.89 × 10^−14^
SLC26A1	4	962,861	997,228	60	6.3491	1.08 × 10^−10^
FBXL19	16	30,924,376	30,970,104	26	6.2787	1.71 × 10^−10^
SMIM15	5	60,443,536	60,468,301	38	6.1459	3.98 × 10^−10^
ERCC8	5	60,159,658	60,250,900	133	5.727	5.11 × 10^−9^
FAM200B	4	15,673,285	15,717,188	66	5.2295	8.50 × 10^−8^
CTF1	16	30,897,928	30,924,881	13	5.2123	9.32 × 10^−8^
ADAM15	1	1.55 × 10^8^	1.55 × 10^8^	24	5.1236	1.50 × 10^−7^
PRSS8	16	31,132,756	31,157,083	17	5.0704	1.99 × 10^−7^
NCOR1	17	15,922,471	16,131,499	206	5.0238	2.53 × 10^−7^
VKORC1	16	31,092,163	31,117,301	10	4.9064	4.64 × 10^−7^
ZNF668	16	31,062,164	31,095,641	19	4.8085	7.61 × 10^−7^
PRSS36	16	31,140,246	31,171,415	19	4.7714	9.15 × 10^−7^
ZNF668	16	31,062,813	31,083,451	14	4.7305	1.12 × 10^−6^
SRCAP	16	30,699,530	30,765,602	18	4.6567	1.61 × 10^−6^
**(c)**						
**SYMBOL**	**CHR**	**START**	**STOP**	**NSNPS**	**ZSTAT**	** *p* **
ZNF165	6	28,038,753	28,067,341	45	7.314	1.30 × 10^−13^
BTN2A2	6	26,373,324	26,405,102	88	6.2064	2.71 × 10^−10^
OR2W1	6	29,001,990	29023,017	5	5.9545	1.30 × 10^−9^
OR12D3	6	29,331,200	29,353,068	6	5.9392	1.43 × 10^−9^
TRMT61A	14	1.04 × 10^8^	1.04 × 10^8^	44	5.9312	1.50 × 10^−9^
OR2J1	6	29,058,386	29,079,658	5	5.6346	8.77 × 10^−9^
HMGN4	6	26,528,633	26,556,482	30	5.5889	1.14 × 10^−8^
OR2B3	6	29,043,985	29,065,090	4	5.2266	8.63 × 10^−8^
BTN3A3	6	26,430,700	26,463,643	54	4.9893	3.03 × 10^−7^
ZNF197	3	44,616,380	44,699,963	70	4.8584	5.92 × 10^−7^
ZNF35	3	44,680,219	44,712,283	18	4.8361	6.62 × 10^−7^
TRIM27	6	28,860,779	28,901,766	11	4.7826	8.65 × 10^−7^
OR2B6	6	27,915,019	27,935,960	14	4.7288	1.13 × 10^−6^
DLST	14	75,338,594	75,380,448	62	4.6985	1.31 × 10^−6^
RHOBTB1	10	62,619,196	62,771,198	238	4.6824	1.42 × 10^−6^
ZNF660	3	44,609,715	44,651,186	46	4.6731	1.48 × 10^−6^
RBM4B	11	66,422,469	66,455,392	19	4.6717	1.49 × 10^−6^
ITGB6	2	1.61 × 10^8^	1.61 × 10^8^	312	4.6636	1.55 × 10^−6^
RBM14-RBM4	11	66,374,097	66,423,940	32	4.5834	2.29 × 10^−6^

Legend: CHR—chromosome; NSNPS—number of SNPs in the gene; START/STOP—start/stop position (bp) of the variant in the genome (GRCh37/hg19 coordinates); ZSTAT—enrichment Z-score statistics; and P—*p*-value.

**Table 3 cells-12-00245-t003:** Single variant association overlap of (a) AD and (b) PD with platelet parameters.

(a)						
SNPs Overlap between	SNP	Gene	Chr:position	Function	Previously Associated with AD	Previously Associated with Platelet Parameters
AD and all platelet parameters	rs6727023	EHD3	2:31475960	Upstream		
	rs62118504	EXOC3L2	19:45734751	Intronic	[23]	
	rs2143926		22:43185180			
	rs4822218	PACSIN2	22:43279611	Intronic		
	rs2267487	PACSIN2	22:43411389	Upstream		
AD, MPV and PDW	rs4575098	B4GALT3	1:161155392	Upstream	[23]	
	rs585021	ITGB5	3:124482869	Intronic		
	rs11620465	LRCH1	13:47250344	Intronic		
	rs3091364		22:43414330			
AD, MPV and Plt	rs12461065	PPP1R37	19:45605308	Intronic	[23]	
	rs12461144	EXOC3L2	19:45723706	Intronic	[23]	
	rs123187		19:45830947		[23]	
AD, PDW and Plt	rs9357551		6:47606029		[24]	Plt [25]
AD and MPV	rs10934680	KALRN	3:124440780	Downstream		
	rs11669338	NECTIN2	19:45382984	Downstream	[24]	
	rs138235833		19:45415285			
	rs620807		19:45706952		[24]	
	rs4803806		19:45708947			
AD and PDW	rs858502	CASTOR3	7:99843353	Intronic	[24]	
	rs7113976		11:85869737		[24]	
	rs283810	NECTIN2	19:45388241	Downstream	[24]	
	rs1160983	TOMM40	19:45397229	Synonymous variant		
	rs584007	APOC1	19:45416478	Upstream	[24]	
	rs117648021	EIF3L	22:38274632	Intronic		
**(b)**						
**SNPs Overlap between**	**SNP**	**Gene**	**Chr:position**	**Function**	**Previously Associated with PD**	**Previously Associated with Platelet Parameters**
PD, MPV and PDW	rs10847839	HIP1R	12:122838013	Intronic		
PD and MPV	rs17689966	CRHR1	17:45833089	Intronic		
	rs9899833	MAPT	17:45915577	Intronic		

No overlaps were found between SNPs associated with MDD and those associated with platelet parameters. Legend: AD—Alzheimer’s disease; PD—Parkinson’s disease; Plt—platelet count; MPV—mean platelet volume; and PDW—platelet distribution width.

**Table 4 cells-12-00245-t004:** Gene enrichment overlap of (a) AD, (b) PD and (c) MDD with platelet parameters.

(a)			
Genes Overlap between	Gene	Previously Associated with AD	Previously Associated with Platelet Parameters
AD and all platelet parameters	AC005779.2		
	AC006126.3		
	CD2AP	[26]	MPV [26] and Plt [26]
	CKM	[16]	
	EHD3		MPV [27], PDW [19] and Plt [27]
	EXOC3L2	[24]	
	KLC3		
	L47234.1		
	MARK4	[24]	MPV [25], PDW [27] and Plt [25]
	PVR	[24]	MPV [27]
AD, MPV and PDW	ADAMTS4	[24]	
	AL590714.1		
	APOA2		
	B4GALT3		
	DEDD	[24]	
	NDUFS2		
	TOMM40L	[28]	
AD, MPV and Plt	GEMIN7	[24]	
AD, PDW and Plt	GATS	[29]	
	AZGP1	[30]	
	PILRA	[24]	
AD and MPV	BLOC1S3	[24]	MPV [25], PDW [27] and Plt [27]
	PPP1R37	[24]	
	PVRL2	[31]	
AD and PDW	APOC1	[23]	PDW [19]
	APOE	[24]	PDW [19] and Plt [27]
	GPC2	[32]	
	PVRIG		
	SLC24A4	[24]	PDW [19]
	STAG3	[33]	
	TOMM40	[24]	
AD and Plt	IGSF23	[24]	
	PICALM	[24]	
**(b)**			
**Genes Overlap between**	**Gene**	**Previously Associated with PD**	**Previously Associated with Platelet Parameters**
PD and MPV	WNT3	[34]	
	AC008498.1		
	SPPL2C	[34]	
	ARHGAP27	[34]	
	CRHR1	[34]	
	MAPT	[34]	Plt [25]
	ELOVL7	[34]	MPV [19]
	ERCC8		
	SMIM15		
	PLEKHM1	[34]	
	KANSL1	[34]	
	NSF	[34]	
	NDUFAF2	[34]	
	STH	[35]	
PD and PDW	SRCAP		MPV [19]
**(c)**			
**Genes Overlap between**	**Gene**	**Previously Associated with AD**	**Previously Associated with Platelet Parameters**
MDD and MPV	OR2J3	[36]	
	TRIM27		
	OR5V1	[37]	
	C11orf31		
	OR2B3		
	NRD1		
	OR12D3		
	HIST1H2BK		
	OR2J1		
	OR2W1		
	HIST1H4K		
	HIST1H2AK		
MDD and PDW	MARK3	[37]	PDW [38]
	TRMT61A		
MDD, PDW and Plt	HIST1H3B		

Legend: AD—Alzheimer’s disease; PD—Parkinson’s disease; MDD—major depressive disorder; Plt—platelet count; MPV—mean platelet volume; and PDW—platelet distribution width.

## Data Availability

AD summary statistics: https://ctg.cncr.nl/software/summary_statistics (accessed on 10 January 2019); PD summary statistics: https://drive.google.com/file/d/1FZ9UL99LAqyWnyNBxxlx6qOUlfAnublN/view?usp=sharing (accessed on 26 March 2020); MDD Summary statistics: https://datashare.ed.ac.uk/handle/10283/3203 (accessed on 30 January 2019); Plt summary statistics: http://ftp.ebi.ac.uk/pub/databases/gwas/summary_statistics/GCST90002001-GCST90003000/GCST90002402/ (accessed on 30 January 2022); MPV summary statistics: http://ftp.ebi.ac.uk/pub/databases/gwas/summary_statistics/GCST90002001-GCST90003000/GCST90002395/ (accessed on 30 January 2022); PDW summary statistics: http://ftp.ebi.ac.uk/pub/databases/gwas/summary_statistics/GCST90002001-GCST90003000/GCST90002401/ (accessed on 30 January 2022).

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
