# Peer review of "A Multi-Trait Association Analysis of Brain Disorders and Platelet Traits Identifies Novel Susceptibility Loci for Major Depression, Alzheimer’s and Parkinson’s Disease"

_cells, 2023, doi:10.3390/cells12020245_

Round 1
Reviewer 1 Report
In the current manuscript, the authors reported a multi-trait association analysis of three neuropsychiatric/neurodegenerative disorders - AD, PD and MDD – and three common platelet parameters, namely Plt, MPV and PDW. By reanalysing the data from previous independent GWAS studies using MTAG, they find novel genome-wide SNP associations with the analysed disorders, not detected in the previous GWAS. Then they checked how the genes located close to the new SNPs are connected through protein-protein interaction, or contribute to pathways.
I found the results of interest with a new list of genes associated with AD, PD and MDD … Unfortunately, I can not judge the method used…
I missed a part of the discussion about the common pathological mechanisms that could be hypothesized from the list of genes that were identified in this study and if it is not indeed a limitation too.
Author Response
In the current manuscript, the authors reported a multi-trait association analysis of three neuropsychiatric/neurodegenerative disorders - AD, PD and MDD – and three common platelet parameters, namely Plt, MPV and PDW. By reanalysing the data from previous independent GWAS studies using MTAG, they find novel genome-wide SNP associations with the analysed disorders, not detected in the previous GWAS. Then they checked how the genes located close to the new SNPs are connected through protein-protein interaction, or contribute to pathways.
I found the results of interest with a new list of genes associated with AD, PD and MDD …Unfortunately, I can not judge the method used…
I missed a part of the discussion about the common pathological mechanisms that could be hypothesized from the list of genes that were identified in this study and if it is not indeed a limitation too.
We thank Reviewer 1 for the comment, which allows us to specify where we report genes with potential pleiotropic roles in two or more of the analyzed disorders. Indeed, in the Discussion we mention some potential genes that may have a role in more than one of these disorders. E.g., “Interestingly, among genes associated with AD, EPPK1, TTLL1, PACSIN2 and TPM4 play a role in the organization of cytoskeletal architecture, that has been identified as an important component in the development of neurodegenerative disorders (36–39). PIF1 prevents telomere elongation by inhibiting the action of telomerase, while ZNF689 and AZGP1P1 are transcription factors involved in cell viability and apoptosis, and both molecular functions can affect cellular aging and the development of age-related disorders such as AD and PD.”
Since the very aim of the study was to identify pleiotropic genes underlying neurodegenerative/neuropsychiatric risk and platelet variability, we do not consider this as a limitation. We hope we clarified these aspects to the reviewer, otherwise please let us know.
Reviewer 2 Report
The review of the manuscript cells-2086795entitled ‘A multi-trait association analysis of brain disorders and platelet traits identifies novel susceptibility loci for major depression, Alzheimer and Parkinson disease’ by Tirozzi et al
Authors conducted a multi-trait analysis of AD, PD, MDD and three platelet parameters using published GWAS data. Results report several novel genes connecting the platelet parameters with the three studied neuropsychiatric disorders.
Remarks:
i) Authors should revise writing the full names of AD and PD in the title (Alzheimer’s and Parkinson’s disease)
ii) When mentioning the URLs for published GWAS summary data, the exact part of the manuscript should be stated. Authors should check the provided URLs because for each one of them the requested page is not found.
iii) Tables used in the manuscript are not completely clear because they are lacking the explanation for used abbreviations.
iv) Please revise the figures’ descriptions. It seems that some of the sentences are duplicated.
v) Authors could consider to include Supplementary Table S2 in the main manuscript.
vi) In the introduction, authors stated ‘Some epidemiological studies consistently reported a positive association between MPV and MDD (7–11), as well as between depressive symptoms and PDW, while evidence of association between Plt and MDD status is less consistent (7)’ and the present study missed to find any connection between platelet parameters and MDD. Authors could discuss and suggest the reasons for this finding.
Author Response
We thank reviewer 2 for the comments. Our replies below.
Remarks:
- i)Authors should revise writing the full names of AD and PD in the title (Alzheimer’s and Parkinson’s disease)
- We revised the name of the disorders in the title.
- ii)When mentioning the URLs for published GWAS summary data, the exact part of the manuscript should be stated. Authors should check the provided URLs because for each one of them the requested page is not found.
- Links for downloading GWAS summary statistics are usually reported in the Data Availability section. We also checked the URLs and replaced those not working with the correct links.
iii) Tables used in the manuscript are not completely clear because they are lacking the explanation for used abbreviations.
- Thanks for the useful suggestion. We added the abbreviations’ legend to all tables.
- iv)Please revise the figures’ descriptions. It seems that some of the sentences are duplicated.
- We checked and corrected the duplicated descriptions (typos). We apologize for the inconvenience.
- v)Authors could consider to include Supplementary Table S2 in the main manuscript.
- Following reviewer’s 2 suggestion, we switched positions of Tables S2 (variant association overlap with platelet parameters) and Table S3 (gene enrichment overlap with platelet parameters) with Table 3 (gene-set enrichments of associations with AD, PD and MDD), now moved to supplementary results.
- vi)In the introduction, authors stated ‘Some epidemiological studies consistently reported a positive association between MPV and MDD (7–11), as well as between depressive symptoms and PDW, while evidence of association between Plt and MDD status is less consistent (7)’ and the present study missed to find any connection between platelet parameters and MDD. Authors could discuss and suggest the reasons for this finding.
To clarify, in the Introduction we referred to previous epidemiological studies and genomic studies, which found less consistent evidence. In the manuscript, we reported associations found through multi-trait association analysis of GWAS summary statistics, which represents the “genomic side” of this relationship. The lack of single variant associations overlaps between MDD and platelet parameters does not necessarily suggest an evidence of no shared genetic underpinnings at all, and may be due to a number of reasons, e.g. the relatively high genetic and phenotypic heterogeneity of MDD and consequent lack of power to detect such associations. Still, it is worth underlining that we observed gene enrichment overlaps between MDD and platelet parameters (see old table S3, now Table 4 in the revised version). In the Strengths and Limitations section we mention a factor potentially hindering to find an overlap between all disorders analyzed and platelet parameters: “…currently only partial evidence of genetic correlation among the disorders and platelet parameters tested exists, which may have hampered the power of the analyses.” Still, it is worth to underline that we observed gene enrichment overlaps between MDD and platelet parameters (see old table S3, now Table 4 in the revised version). In the same line, we added the following sentence to the Discussion section “Still, further studies are needed to clarify the variant associations overlap between platelet parameters and MDD, which we were not able to identify here, possibly due to the genetic and phenotypic heterogeneity of depression.”
Reviewer 3 Report
Dear Dr.,
Title: A multi-trait association analysis of brain disorders and platelet traits identifies novel susceptibility loci for major depression, Alzheimer and Parkinson's disease
Manuscript ID: cells-2086795
Overall comments: Authors described in this manuscript: the role of platelet traits mediated novel susceptibility of brain loci in major depression, Alzheimer and Parkinson's disorders via gene-based enrichment tests. The overall manuscript is well written and it has novelty in this field of research. However, some of the contents need to improve for the better quality of this manuscript. It can be accepted after minor revision.
Specific comments:
1. The abstract section is well written.
2. The introduction and materials & methods sections are well framed.
3. Figures 1 and 2 are blurred and the text font can increase. Need to improve the quality of the image.
4. Figure 3 text is not readable. Need to improve.
5. Discussion section is too lengthy. Need to rewrite with relevant concise statements.
6. In reference sections: All the references are placed as recent and relevant references.
Author Response
Specific comments:
- The abstract section is well written.
- The introduction and materials & methods sections are well framed.
- Figures 1 and 2 are blurred and the text font can increase. Need to improve the quality of the image.
- We improved the quality of the figures.
- Figure 3 text is not readable. Need to improve.
- Figure 3 was also improved.
- Discussion section is too lengthy. Need to rewrite with relevant concise statements.
- According to reviewer’s 3 suggestion, we shrinked the section and rephrased some statements to make them more concise.
- In reference sections: All the references are placed as recent and relevant references.
- We are not sure we got the objection by reviewer 3. If this refers to the number of references, we attempted to reduce it, as well, trying to leave the most relevant ones. If he/she refers to reference 57, this was erroneously left in the Discussion, hence removed.